# Whole-Body Vibration Prevents Neuronal, Neurochemical, and Behavioral Effects of Morphine Withdrawal in a Rat Model

**DOI:** 10.3390/ijms241814147

**Published:** 2023-09-15

**Authors:** Gavin C. Jones, Christina A. Small, Dallin Z. Otteson, Caylor W. Hafen, Jacob T. Breinholt, Paul D. Flora, Matthew D. Burris, David W. Sant, Tysum R. Ruchti, Jordan T. Yorgason, Scott C. Steffensen, Kyle B. Bills

**Affiliations:** 1Neuroscience Center, Brigham Young University, Provo, UT 84602, USA; 2Department of Biomedical Sciences, Noorda College of Osteopathic Medicine, Provo, UT 84606, USA

**Keywords:** opioid abuse, addiction, morphine, VTA, nucleus accumbens, dopamine, whole-body vibration, mechanoreceptors

## Abstract

Peripheral mechanoreceptor-based treatments such as acupuncture and chiropractic manipulation have shown success in modulating the mesolimbic dopamine (DA) system originating in the ventral tegmental area (VTA) of the midbrain and projecting to the nucleus accumbens (NAc) of the striatum. We have previously shown that mechanoreceptor activation via whole-body vibration (WBV) ameliorates neuronal and behavioral effects of chronic ethanol exposure. In this study, we employ a similar paradigm to assess the efficacy of WBV as a preventative measure of neuronal and behavioral effects of morphine withdrawal in a Wistar rat model. We demonstrate that concurrent administration of WBV at 80 Hz with morphine over a 5-day period significantly reduced adaptations in VTA GABA neuronal activity and NAc DA release and modulated expression of δ-opioid receptors (DORs) on NAc cholinergic interneurons (CINs) during withdrawal. We also observed a reduction in behavior typically associated with opioid withdrawal. WBV represents a promising adjunct to current intervention for opioid use disorder (OUD) and should be examined translationally in humans.

## 1. Introduction

Over the past decade, opioid use disorder (OUD) has reached epidemic proportions, in part driven by inappropriate use of prescription opioid pain killers [1]. OUD was identified as a public health emergency in August 2017 by the US President [2]. Nearly 3 million Americans have OUD and less than approximately 20% receive treatment [3]. The current FDA-approved treatments for OUD include the opioid receptor (OR) agonists methadone and buprenorphine, the μ-OR (MOR) antagonist naltrexone [4], and various forms of psychosocial intervention [5]. Stress and anxiety associated with OUD are major factors for initial drug seeking and relapse from chronic abuse, which suggests a possible role for anxiolytic approaches in the prevention and treatment of OUD [6,7,8,9].

Both animal and human studies indicate that key elements within the mesocorticolimbic dopamine (DA) system, including the ventral tegmental area (VTA) and nucleus accumbens (NAc), constitute major brain substrates underlying the reinforcing effects of drugs of abuse, including opioids. Dopamine is the canonical neurotransmitter implicated in motivated behavior and reward learning. Current dogma maintains that DA neuron activation and release in the mesocorticolimbic DA system originating in the midbrain VTA and projecting to the NAc and other limbic structures is a scalar index of reward [10]. This system is known to be involved in reward stemming from natural behaviors such as feeding [11,12,13] and drinking [14,15], as well as other rewards such as intracranial self-stimulation [16,17,18]. It has also been implicated in the habit-forming actions of several addictive drugs [19], including opioids. The dogma is that any drug or behavior that increases mesolimbic DA neuron activity and release will be reinforcing and potentially addictive [20,21,22].

Although the prevailing dogma is that DA neurons mediate the rewarding and addictive properties of drugs of abuse [10], VTA GABA neurons have garnered much interest for their role in modulating DA release and perhaps as independent substrates mediating reward or aversion [23,24,25,26,27,28,29]. We have shown previously that acute administration of ethanol, opioids, or cocaine inhibits VTA GABA neurons [23,24,25,26,30], leading to a net disinhibition of VTA DA neurons [31,32,33]. In contrast, during EtOH or opioid withdrawal, VTA GABA neurons become hyperactive [23,34], leading to decreased mesolimbic DA activity and DA release in the NAc [19,35,36,37,38]. This reduction in mesolimbic DA transmission is theorized to be the primary driver of relapse [39]. However, VTA GABA neurons appear to be the primary adaptive substrate in the VTA for drug dependence, as we have shown in multiple reports that GABA_A_ receptors (GABA_A_R) switch their function following alcohol and opioid withdrawal [40,41,42,43,44].

Peripheral mechanoreceptor-based treatments have previously been shown to modulate the brain substrates underlying addiction. For example, acupuncture has been shown to modulate VTA GABA neurons via somatosensory pathways and reduce alcohol, cocaine, and methamphetamine psychomotor effects and drug-seeking behavior [45,46,47,48,49,50]. Mechanical stimulation at HT7 (Shenmen acupoint) activates the dorsal column medial lemniscus (DCML) pathway, synapses in the nucleus cuneatus (CN), and subsequently relays through the thalamus and lateral habenula [51] before finally modulating the excitability of VTA GABA neurons [45]. The effects of HT7 acupoint stimulation can be attenuated with ablation of the DCML and are mediated by endogenous opioids [45,47,52], presumably through δ-opioid receptors (DORs) that are expressed on GABA neurons in, or projections of GABA neurons to the NAc [53]. Pertinent to addressing drug relapse, such treatments have also been shown to modulate behavioral symptoms of stress, anxiety, and depression [54,55,56].

We have previously reported that stimulation of mechanoreceptors in the cervical spine (MStim), an area with a relatively high concentration of primary mechanoreceptors [57], for 1–2 min at 40 and 80 Hz inhibits VTA GABA neuron firing, enhances VTA DA neuron firing, and increases DA release in the NAc [58]. VTA effects are driven by cholinergic interneurons (CINs) and DOR number and translocation in the NAc [59]. Furthermore, we have shown that whole-body vibration (WBV) at 80 Hz reduces adaptations in VTA GABA neurons and DA release in response to a reinstatement dose (2.5 g/kg) of ethanol (EtOH) [60]. Behavioral indices of EtOH withdrawal (e.g., anxiety) were also substantively ameliorated with concurrent application of WBV, and WBV significantly increased the overall frequency of ultrasonic vocalizations, suggesting an increased positive affective state [61].

In this study, we investigate the role of WBV as a simulation of mechanoreceptor-based treatment modalities. We evaluate its ability to prevent morphine-induced changes to VTA GABA neuron firing and DA release in the NAc as we have previously done with EtOH. We likewise examine its effect on behavioral indices related to morphine withdrawal. As opioids and EtOH have similar effects on VTA neurons and DA release, we hypothesized that WBV would be an adequate stimulus to block morphine-induced desensitization of VTA GABA neurons and changes in NAc DA release in response to acute morphine reinstatement. We further hypothesized that WBV would block anxiety-related behaviors normally associated with withdrawal from opioid treatment.

## 2. Results

### 2.1. Effect of WBV on Morphine-Induced Changes to VTA GABA Neurons

We have previously demonstrated that VTA GABA neurons develop tolerance to an acute opioid challenge and enhanced baseline firing rate during opioid withdrawal [43]. As WBV reduces EtOH effects on VTA GABA neurons [60], we hypothesized that WBV would similarly reduce morphine effects on VTA GABA neurons. We recorded VTA GABA neuron firing rate in vivo under isoflurane anesthesia in rats that had undergone five days of saline or morphine injections with or without concurrent WBV treatment (Figure 1). Morphine-dependent rats were allowed to enter a withdrawal state (24 h) before experimentation. The GABA neuron firing rate was recorded before and after a reinstatement dose of morphine (1 mg/kg IP). The firing rate 30-60 min post-injection taken as a percentage of baseline rate was significantly higher in morphine-control rats (94.8 ± 8.5%) than all other groups (*F*_3,67_ = 34.6, *p* < 0.0001; Figure 1C,D). Importantly, morphine + WBV rats demonstrated a post-injection firing rate similar to that of saline rats (*p* > 0.05). These findings suggest that when administered concurrent with morphine, WBV blocks the desensitization of VTA GABA neurons to reinstatement exposure.

### 2.2. Effect of WBV on Morphine-Induced Changes to NAc DA Release

As WBV has previously been shown to decrease EtOH-induced changes to NAc DA release [60], WBV effects on morphine-induced changes to NAc DA release were next observed. Dialysate samples were collected from the NAc before and after a 1 mg/kg morphine challenge in isoflurane-anesthetized rats (Figure 2). Following a 24-h withdrawal state, the average DA concentration in saline-treated animals exhibited a significant morphine-induced rise in DA levels as compared to baseline during the first hour post-injection (saline-control, 144 ± 9.02% baseline, *t*_16_ = 4.7254, *p* = 0.0002; saline + WBV, 136 ± 7.07% baseline, *t*_10_ = 4.9399, *p* = 0.0006; Figure 2B). Similar to saline-treated groups, morphine + WBV rats also experienced a significant rise in DA level (150 ± 7.53% baseline, *t*_14_ = 6.6882, *p* < 0.0001) and did not significantly differ from saline-treated groups (*p* > 0.05). Morphine-control animals had significantly lower increases in DA concentration than all other groups during the first (*F*_3,57_ = 12.55, *p* < 0.0001) and second (*F*_3,58_ = 3.804, *p* = 0.0147) hours post-injection. There were significant differences between morphine + WBV and morphine-control animals at 20 min (*p* = 0.0443), 40 min (*p* = 0.0029), and 60 min (*p* = 0.0213), as well as 120 min (*p* = 0.0495) and 180 min (*p* = 0.0083). Interestingly, while saline groups trended down towards baseline DA levels during hour 2 (saline-control, 114 ± 10.2% baseline; saline-WBV, 118 ± 11.4% baseline; Figure 2A) and hour 3 (saline-control, 104 ± 4.97% baseline; saline-WBV, 90.1 ± 7.72% baseline; Figure 2A), morphine + WBV animals remained significantly elevated above all other animals during hour 3 (127 ± 4.96% baseline; *F*_3,54_ = 8.495, *p* < 0.0001). Taken altogether, these results suggest that WBV blocks morphine-induced changes to NAc DA neural activity.

### 2.3. Effect of WBV on Morphine-Induced Changes in NAc CIN DOR Expression

As mechanical stimulation targeted at cervical spine has previously shown to modulate DOR expression in NAc CINs [59], we next evaluated changes in DOR expression in NAc CINs induced by WBV with or without sub-chronic morphine administration via immunohistochemistry. DOR expression, as measured by mean fluorescence intensity (MFI), in morphine rats without WBV was significantly higher than all other groups (3.30 ± 0.384 DOR MFI; *F*_3,82_ = 13.38, *p* < 0.0001; Figure 3B). Morphine + WBV rats did not significantly differ from rats that did not receive drug, independent of WBV status (*p* > 0.05). MANOVA revealed a significant effect from drug status (morphine vs. no drug; *F*_1,82_ = 25.201, *p* < 0.0001), treatment status (WBV vs. control; *F*_1,82_ = 5.525, *p* = 0.0212), as well as a significant interaction between drug and treatment (*F*_1,82_ = 9.424, *p* = 0.0029). This suggests that the effect of WBV on DOR expression varies somewhat in the presence of morphine, but that WBV alone has a significant effect.

When taken as the ratio of DOR MFI in the membrane vs. cell cytosol, no drug + WBV rats showed significantly higher expression than all other groups (1.80 ± 0.0663; *F*_3,82_ = 31.42, *p* < 0.0001; Figure 3C). Morphine + WBV rats showed decreased expression compared to no drug-control rats (morphine + WBV, 1.18 ± 0.384; no drug-control, 1.38 ± 0.0618; *p* = 0.0205) and did not significantly differ from morphine + no WBV rats (1.24 ± 0.384; *p* > 0.05). As before, MANOVA revealed a significant effect from drug status (morphine vs. no drug-control; *F*_1,82_ = 54.07, *p* < 0.0001), treatment status (WBV vs. no treatment; *F*_1,82_ = 17.47, *p* < 0.0001), and a significant interaction between drug and treatment (*F*_1,82_ = 22.72, *p* < 0.0001), again suggesting that WBV has a significant effect on DOR expression, which is affected by the co-administration of morphine.

### 2.4. Effect of WBV on Anxiety Induced by Morphine Withdrawal

We next examined the effect of WBV on anxiety related to opioid withdrawal, a common factor in opioid use relapse [7,8,9]. As we have previously shown that WBV reduces anxiety associated with EtOH withdrawal [60], we hypothesized that WBV would also reduce anxiety associated with morphine withdrawal. The relative level of anxiety displayed in rats was measured via the elevated-plus maze [62] before rats received any drug or treatment (“baseline”), after 5 days of drug and treatment administration (“intoxicated”), and 24-h after the final drug administration wherein morphine-dependent rats had entered a withdrawal state (“withdrawal”). Baseline measurements showed no significant difference between experimental groups (*F*_3,48_ = 0.863, *p* > 0.05; Figure 4A). After five days of morphine injections, morphine-receiving rats spent significantly less time in the open arm than saline-receiving rats, independent of WBV treatment (MANOVA; morphine vs. saline, *F*_1,48_ = 18.872, *p* < 0.0001; WBV vs. control, *F*_1,48_ = 0.196 *p* > 0.05; Figure 4B). During the withdrawal period, morphine-control rats spent significantly less time in the open arm than all other groups (120 ± 8.31 s; *F*_3,48_ = 6.082, *p* = 0.00136) while morphine + WBV rats did not significantly differ from saline-control rats (morphine + WBV, 151 ± 11.1 s; saline-control, 162 ± 14.3 s; *p* > 0.05; Figure 4C).

Within-group paired comparison shows that morphine-control rats spent significantly less time in the open arm during withdrawal than during baseline measurements (baseline, 165 ± 12.8 s; paired *t*_14_ = 3.2726, *p* = 0.00556). Morphine + WBV rats did not show a similar decline from baseline (baseline, 173 ± 12.7 s; paired *t*_13_ = 1.4757, *p* > 0.05). Interestingly, while saline-control rats showed an initial increase of open time on day 5 with a restoration to baseline levels on day 6, saline + WBV animals remained significantly more elevated than baseline levels (baseline, 142 ± 17.1 s; day 6, 185 ± 9.67 s; *t*_10_ = −2.2403, *p* = 0.0498) and were significantly higher than morphine + WBV animals on day 6 (*p* = 0.03912). During withdrawal, MANOVA analysis reveals a significant effect from treatment status (WBV vs. control; *F*_1,48_ = 6.175, *p* = 0.0165), drug status (morphine vs. saline; *F*_1,48_ = 11.927, *p* = 0.00117), but not the interaction between drug and treatment (*F*_1,48_ = 0.143, *p* > 0.05). Taken altogether, these results suggest an anxiolytic effect of WBV within morphine withdrawal but not during intoxication. The anxiolytic effect of WBV may also be present independent of drug intake, as shown in saline animals treated concurrently with WBV.

## 3. Discussion

We have previously reported that 80 Hz mechanical stimulation (MStim) acts on the NAc to increase local DA release and that this effect is mediated by activation of cholinergic neurons and DORs [58,59]. Further, projections from the NAc then cause a depression in VTA GABA neuron firing, which results in VTA DA neuron disinhibition and a subsequent increase in firing. We have recently reported that WBV at frequencies shown in these previous studies to modify VTA neuronal activity and enhance DA release reduces neuronal and behavioral indices of withdrawal from EtOH, including anxiety [60]. The current study was designed to investigate if these neuromodulatory changes by WBV are sufficient to alter sub-chronic opioid effects on VTA GABA neurons, DA release, DOR expression on CINs in the NAc, and anxiety in the elevated-plus maze procedure.

The effects of 80 Hz WBV treatment, given twice daily for 15 min, concurrently with dependence-inducing intermittent morphine injections, were tested on various measures of morphine withdrawal. While acute administration of morphine reduces VTA GABA neuron firing rate, this effect has been shown to desensitize with chronic exposure [34]. Interestingly, concurrent administration of 80 Hz WBV with sub-chronic morphine blocks morphine-induced desensitization (Figure 1C,D). Mechanical stimulation has been previously shown to increase DA levels in the NAc 2 h post-MStim [59]. This effect activates medium spiny neurons that project back to the VTA and ultimately influence VTA GABA neuron firing [58,63]. This feedback mechanism could be responsible for the described WBV effects.

Dopamine levels were also measured following withdrawal and reinstatement. Morphine naïve and exposed animals demonstrated characteristic responses following 1 mg/kg morphine injection with respective increases and maintenance of baseline [64,65]. However, rats that received WBV concurrent with morphine injections exhibited increases in DA levels akin to rats in a morphine naïve state (Figure 2A). Interestingly, these rats not only experienced acute increases in DA levels in-line with naïve rats, but those levels remained higher than all other groups by the third hour of testing, suggesting a lasting impact of the interaction between morphine and WBV (Figure 2D). Morphine administration has been shown to increase expression levels of DORs while decreasing expression of MORs [66,67,68]. Additionally, we have previously shown that targeted mechanical stimulation of cervical spine at 80 Hz produces increased translocation of DORs to NAc neuron membranes [59]. This commonality between morphine and mechanical stimulation could contribute to the altered release pattern of DA in rats exposed to sub-chronic morphine and WBV.

To evaluate the influence of DOR translocation induced by WBV, immunohistochemical analysis was performed to quantify DOR expression on CINs in the NAc. The experiments revealed a significant increase in MFI from DORs expressed on CINs in morphine-treated rats when compared to all other groups (Figure 3B), and increased DOR translocation to the cellular membrane of CINs in rats that were treated with WBV alone (Figure 3C). The latter finding replicates previous studies that have shown mechanical stimulation producing this effect in isolation [59]. Further, WBV in sub-chronic morphine rats demonstrates no significant change in DOR MFI when compared with morphine-naïve rats and is suggestive of a protective effect when compared to morphine administration alone (Figure 3A,B).

Finally, we hypothesized that WBV effects on VTA GABA neurons, DA release, and its ability to ameliorate measurements of anxiety are due to its induction of DA release in the NAc. The ascending mesolimbic cholinergic system initiates negative emotional states, and the ascending DA system initiates positive emotional states [69,70]. Importantly, WBV rescued morphine induced reductions in open arm time during an elevated plus behavioral assay, which is a proxy for anxiety (Figure 4) [62]. It is noteworthy that prior reception of WBV was not sufficient to alter behavioral outcomes in acute intoxication.

This study adds to the mounting evidence that peripheral mechanoreceptor activation holds promise as a potential therapeutic neuromodulatory modality because of its ability to alter neuron firing patterns and induce DA release. Specifically, it is suggestive that for conditions characterized by reductions in DA levels, targeted WBV should be considered for future studies to evaluate its efficacy. This is particularly important because of its low risk-benefit profile, non-invasive nature, and cost effectiveness. It is noteworthy that WBV shares potential mechanistic underpinnings with several physical therapeutics including osteopathic and chiropractic manipulation as well as acupuncture. This study was limited to examining the ability of WBV to reduce neuronal, neurochemical, and behavioral indices of withdrawal when administered concomitantly with sub-chronic morphine. Future studies should address whether WBV can reverse these changes after withdrawal is already established. Future studies should also evaluate effectiveness in a human model and could include studies into the mesolimbic effects of these varied physical modalities.

## 4. Materials and Methods

### 4.1. Animal Subjects

Male Wistar rats (350–500 g, 70–120 PND, *n* = 83) were used for all experimentation. Initial testing with male and female rats demonstrated no significant difference in WBV-induced effects on mesolimbic neurons. Once weaned at PND 21, animals were housed in groups of 2–3 and given access to solid food and water ad libitum. Animals were housed at a fixed temperature (21–23 °C) and humidity (55–65%) and placed on a reverse light/dark cycle with lights on from 10 p.m. to 10 a.m. For each methodology employed, animals were treated in strict accordance with the Brigham Young University (BYU) Animal Research Committee (IACUC) guidelines, which incorporate and exceed current NIH guidelines. The BYU IACUC reviewed and approved the procedures detailed herein.

### 4.2. Whole-Body Vibration Apparatus

Animals were placed in a sound-attenuating chamber (57 cm × 57 cm × 50.8 cm), with floors constructed of 1 cm thick aluminum plate and isolated from the walls of the chamber with 4 vibration isolators at each corner. A low frequency effect (LFE) audio transducer (miniLFE ButtKicker, The Guitammer Company, Westerville, OH, USA) was suspended below the center of the floor. An 80 Hz, 500 mV pp sine wave was generated using a Wavetex Datron Universal Waveform Generator model 195 (San Diego, CA, USA) and amplified using a Crown model XLi 3500 (Los Angeles, CA, USA) amplifier. The vibration acceleration on the plate was 1.86 m/s^2^. Parameters were selected congruent with previous studies [59,60].

### 4.3. Morphine Administration and Whole-Body Vibration

Rats were randomly divided into four groups based on drug administered and treatment condition as follows: saline + no treatment, *n* = 20; saline + 80 Hz WBV, *n* = 20; morphine + no treatment, *n* = 22; morphine + 80 Hz WBV, *n* = 21. Over a 5-day period, each animal received twice-daily treatment (9 a.m. and 5 p.m.) for 15 min, followed by a subcutaneous injection of the animal’s assigned drug. Animals assigned to the “no treatment” condition were placed in an unpowered WBV apparatus for 15 min. Injected morphine concentrations increased incrementally from 10 mg/kg to 20 mg/kg over the 5-day period. On the final day of injections, rats were administered their assigned drug at 9 a.m. but not at 5 p.m.to facilitate a 24-h drug abstinence period and thereby elucidate a natural withdrawal state in morphine-receiving rats beginning at 9 a.m. the following day. Rats received their assigned treatment (WBV or no treatment) at 9 a.m. and 5 p.m.on the final day of injections. Evaluative signs of withdrawal included wet dog shakes, teeth chattering, mastication, and eye twitch [71].

### 4.4. In Vivo Single Cell Recordings and Characterization of VTA GABA Neurons

Rats (*n* = 30) were anesthetized using isoflurane and placed in a stereotaxic apparatus. Anesthesia was maintained at 1.5% with 2.0 L/min of air flow from a nebulizer driven by an oxygen concentrator. Body temperature was maintained at 37.0 ± 0.4 °C by a feedback-regulated heating pad. With the skull exposed, a hole was drilled for placement of a 3.0 M KCl-filled micropipette (2 to 4 MΩ; 1–2 µm inside diameter), driven into the VTA with a piezoelectric microdrive (EXFO Burleigh 8200 controller and Inchworm, Victor, NY, USA) based on the following stereotaxic coordinates: from bregma: 5.6 to 6.5 mm posterior, 0.5 to 1.0 mm lateral, 6.5 to 9.0 mm ventral. Potentials were amplified with an Axon Instruments Multiclamp 700A amplifier (Union City, CA, USA). Single-cell activity was filtered at 0.3 to 10 kHz (3 dB) with the Multiclamp 700A amplifier and displayed on Tektronix (Beaverton, OR, USA) digital oscilloscopes. Potentials were sampled at 20 kHz (12-bit resolution) with National Instruments (Austin, TX, USA) data acquisition boards in Macintosh computers (Apple Computer, Cupertino, CA, USA). Extracellularly recorded action potentials were discriminated with a World Precision Instruments WP-121 Spike Discriminator (Sarasota, FL, USA) and converted to computer-level pulses. Single-unit potentials and discriminated spikes were captured by National Instruments NB-MIO-16 digital I/O and counter/timer data acquisition boards in Macintosh PowerPC computers.

VTA GABA neurons were identified by previously established stereotaxic coordinates and by spontaneous electrophysiological and pharmacological criteria [72]. VTA GABA neuron discharge activity characteristics included a relatively fast firing rate (>10 Hz), ON-OFF phasic non-bursting activity, and an initially negative spike with a duration less than 200 µs. We evaluated only the spikes that had a greater than 5:1 signal-to-noise ratio. After positive neuron identification, the baseline firing rate was measured for 10 min to ensure stability. Rats were then administered an IP injection of saline and neuron activity was recorded for another 10 min. A reinstatement dose of morphine (1 mg/kg IP) was then administered, and activity was recorded for several hours or until GABA neuron activity diminished to an unobservable level.

### 4.5. In Vivo Microdialysis and High-Performance Liquid Chromatography of NAc DA

Rats (*n* = 42) were anesthetized using isoflurane and placed in a stereotaxic apparatus. Anesthesia was maintained at 1.5% with 2.0 L/min of air flow from a nebulizer driven by an oxygen concentrator. Body temperature was maintained at 37.0 ± 0.4 °C by a feedback-regulated heating pad. With the skull exposed, microdialysis probes (MD-2200, BASi Research Products, West Lafayette, IN, USA) were inserted into the NAc based on stereotaxic coordinates (from bregma: 1.6 mm anterior, 1.9 mm lateral, 8.0 mm ventral). Artificial cerebrospinal fluid (aCSF) composed of 150 mM NaCl, 3 mM KCl, 1.4 mM CaCl_2_, and 0.8 mM MgCl_2_ in 10 mM phosphate buffer was perfused through the probe at a rate of 3.0 µL/min. After a 24-h withdrawal period, samples were collected every 20 min for 4 h with a reinstatement dose of morphine (1 mg/kg IP) occurring after the first hour had elapsed. Baseline was determined as an average of samples collected for 1 h prior to reinstatement dose. Dopamine concentration in dialysate samples was evaluated using a high-performance liquid chromatography (HPLC) pump (Ultimate 3000, Dionex, Sunnyvale, CA, USA) connected to an electrochemical detector (Coulochem III, ESA, Thermo Fisher Scientific, Waltham, MA, USA). The coulometric detector included a guard cell (5020, ESA) set at +275 mV, a screen electrode (5014B, ESA) set at −100 mV, and a detection electrode (5014B, ESA) set at +220 mV. DA was separated using a C18 reverse phase column (HR-80, Thermo Fisher Scientific, Waltham, MA, USA). Mobile phase containing 150 mM NaH_2_PO_4_, 4.76 mM anhydrous citric acid, 3 mM sodium dodecyl sulfate (SDS), 50 µM ethylenediaminetetraacetic acid, 15% *v*/*v* acetonitrile and 10% *v*/*v* methanol (final pH = 5.6 by NaOH) was pumped through the system at a flow rate of 0.4 mL/min. External DA standards (1 nM and 10 nM) were assayed concurrently with the samples to allow for construction of a calibration curve using Chromeleon software (v7.2, Thermo Fisher Scientific, Waltham, MA, USA), which was used to estimate the DA concentration of the dialysate samples.

### 4.6. Immunohistochemistry and Confocal Microscopy

Rats (*n* = 12) were anesthetized in 2–4% isoflurane and perfused with 1× PBS composed of 137 mM NaCl, 3 mM KCl, 10.4 mM NaH_2_PO_4_, and 1.8 mM NaHCO_3_ followed by 4% paraformaldehyde (PFA). Brains were then extracted and held in 4% PFA for 24–48 h after which the brains were transferred to a 30% sucrose solution. The brains were later flash-frozen in dry ice, sliced in 30 µm sections, and incubated in 1× PBS. The immunohistochemistry and fluorescence imaging of cholinergic neurons in the NAc were visualized using an anti-choline acetyltransferase antibody (Millipore, CAT# AB144P). Antibody labeling protocol was used to label delta opioid receptors (DORs; Invitrogen, # PA5-86357, Carlsbad, CA, USA). The anti-DOR antibody was conjugated using a CF 405 labeling kit (Biotium, CAT#92232, Fremont, CA, USA). Slices were then incubated with 1:50 dilution of antibody in PBS-T (PBS with 0.4% Triton X-100) for 2 h at room temperature and subsequently washed with 1× PBS-T 3 times for 5 min each. Coverslips were mounted onto glass microscope slides using mounting media (Fluoromount G, ThermoFisher, Waltham, MA, USA). Images were captured using an Olympus Fluoview FV1000 microscope using a 60× oil 1.5 NA objective (Olympus, Tokyo, Japan). CF 405 and CF 488 fluorophores were excited with 405 and 488 nm lasers, respectively. Mean fluorescence intensity was analyzed using Fiji v20221201-1017 [73].

### 4.7. Elevated-Plus Maze

The elevated-plus maze (EPM) consisted of two open arms (56 cm × 10 cm) and two closed arms (56 cm × 10 cm with 15 cm walls). The walls of the closed arms were covered with a roof (56 cm × 10 cm). The maze was suspended off the ground by 61 cm. Foam padding was placed under the maze to protect from falls. Rats were handled and allowed to habituate to the testing room over 3 sessions lasting 20–30 min each. Careful measures were taken to ensure consistency in rodent handling and room environment including light levels, temperature, external noise volume, relative placement of equipment, etc. On the testing days, rats (*n* = 55) were placed in the center of the apparatus and allowed to move freely for 5 min while being video recorded. Recordings were taken on 3 occasions: (1) before the animal was administered a drug or WBV (baseline measurement); (2) after 5 days of drug and WBV administration; and (3) after a 24-h drug abstinence period. Recordings were later evaluated by two independent reviewers for time spent in each section of the maze. Two recordings were thrown out because the rats fell from the platform. Time spent in open versus closed arms was used as a measure of anxiety level where more time spent in the closed arm is correlated with higher levels of anxiety [57].

### 4.8. Statistical Analyses

For single-unit electrophysiology experiments, discriminated spikes were analyzed with IGOR Pro software (v8.04, Wavemetrics, Lake Oswego, OR, USA). Experimental group firing rates were averaged across bins and then compared using a one-way analysis of variance test (ANOVA) followed by Tukey HSD post-hoc test. Where appropriate, within-group comparison over time utilized a Paired *t*-test. Between-group comparison of averaged bins during key time points utilized a Welch two-sample *t*-test. Average depression or excitation in firing was calculated from 10 to 40 min post injection. The baseline firing rate was calculated from the average of the final 60 s of firing rate data after an injection of saline equal to the volume of morphine challenge injection and after 10 min of recording to ensure neuron stability. The results from all groups were derived from calculations performed on ratemeter records and expressed as means and percent of baseline firing rate ± SEM.

For microdialysis experiments, the area under the curve for the DA peak was extracted and a two-point calibration was used to approximate the DA concentration. All collections were normalized to the first hour of baseline collection before injection occurred. DA release for each time point was expressed as a percentage of baseline ± SEM and was compared using a one-way ANOVA followed by Tukey HSD post-hoc test. For binned comparisons, values were averaged over the specified time frame and compared similarly. Where appropriate, within-group comparison over time utilized a Paired *t*-test. Between-group comparison of averaged bins during key time points utilized a Welch two-sample *t*-test.

For immunohistochemistry, MFI was calculated as the ratio of cell MFI divided by background MFI. The ratio of membrane MFI to cytoplasmic MFI was also calculated. Results between groups were compared using a one-way ANOVA followed by Tukey HSD post-hoc test. A multivariate ANOVA (MANOVA) was used to analyze the effect of each variable separately as well as possible interactions between variables.

For behavioral experiments, after blinded scoring, results between groups were compared for each day using a one-way ANOVA followed by Tukey HSD post-hoc test. Where appropriate, within-group comparison between days utilized a Paired *t*-test. Between-group comparison of measures on the same day utilized a Welch two-sample *t*-test. A multivariate ANOVA (MANOVA) was used to analyze the effect of each variable separately as well as possible interactions between variables.

All analyses were conducted in RStudio (R v4.2.3, R Core Team, 2023; RStudio v2023.03.15, Posit team, 2023) and IGOR Pro (v 8.04, Wavemetrics, Oswego, OR, USA). All group values are reported as means ± SEM. On graphs, significance levels for within-group comparisons are indicated with asterisks (*, **, ***), and between-group comparisons are indicated with hashes (#, ##, ###), corresponding to levels *p* < 0.05, 0.01 and 0.001, respectively. The figures were constructed with IGOR Pro.

## Figures and Tables

**Figure 1 ijms-24-14147-f001:**
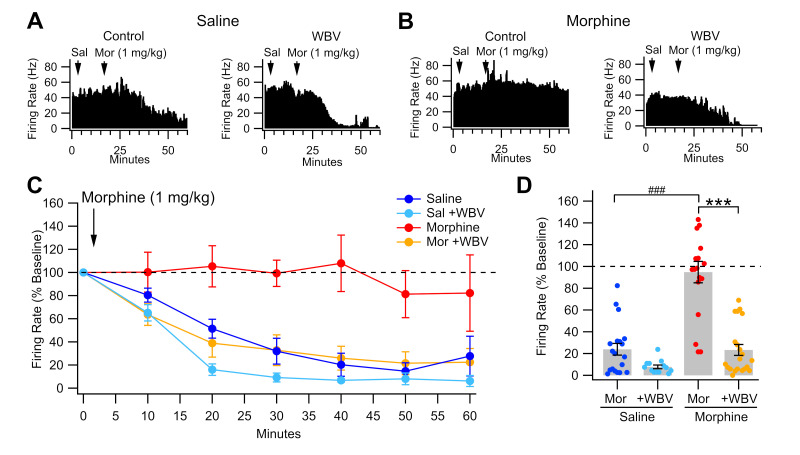
Effects of WBV on VTA GABA neuron firing rate in response to morphine reinstatement during withdrawal. (**A**,**B**) Representative ratemeter records for GABA neuron response to morphine after five days of saline (**A**) or morphine (**B**) with or without WBV. Arrows denote timing of saline or acute morphine challenge (1 mg/kg) administration. Note that acute morphine challenge inhibits the VTA GABA neuron firing rate after saline administration but has little effect after sub-chronic morphine exposure. WBV restores the ability of morphine to inhibit VTA GABA neuron firing rate. (**C**) Time course of all groups’ post-injection firing rate taken as a percentage of baseline firing rate. (**D**) Summary demonstrating that WBV significantly reduces morphine-induced desensitization of GABA neurons to morphine reinstatement. Significance levels for comparisons within a drug condition are indicated with asterisks (***, *p* < 0.001) and for between drug conditions with hashes (###, *p* < 0.001). Not all comparisons are included.

**Figure 2 ijms-24-14147-f002:**
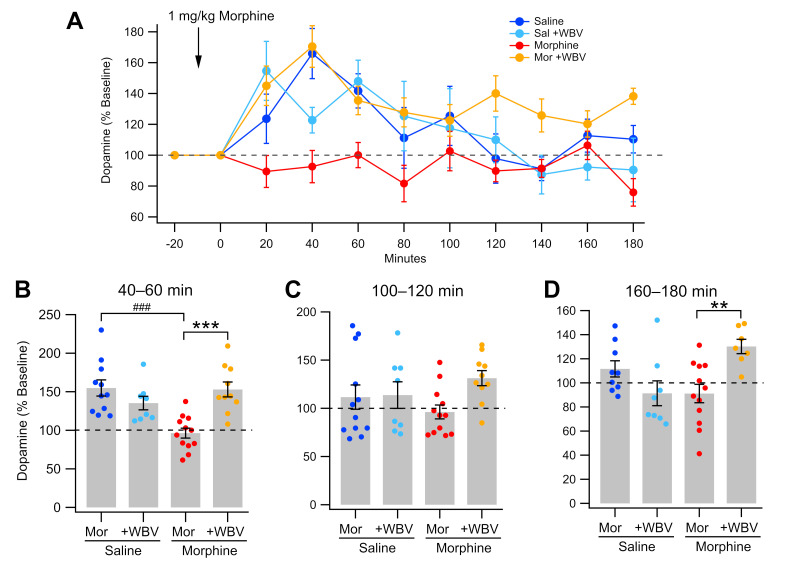
Effects of WBV on DA release in the NAc during morphine withdrawal. (**A**) Summarized time course of morphine enhancement of basal DA release in the NAc. Note that saline treated animals show distinct differences when compared to animals that received morphine or WBV. (**B**–**D**) Comparison of DA release at time intervals: 40–60 min; 100–120 min; and 160–180 min after morphine injection. Significance levels for comparisons within a drug condition are indicated with asterisks (**, *p* < 0.01; ***, *p* < 0.001) and for between drug conditions with hashes (###, *p* < 0.001). Not all comparisons are included.

**Figure 3 ijms-24-14147-f003:**
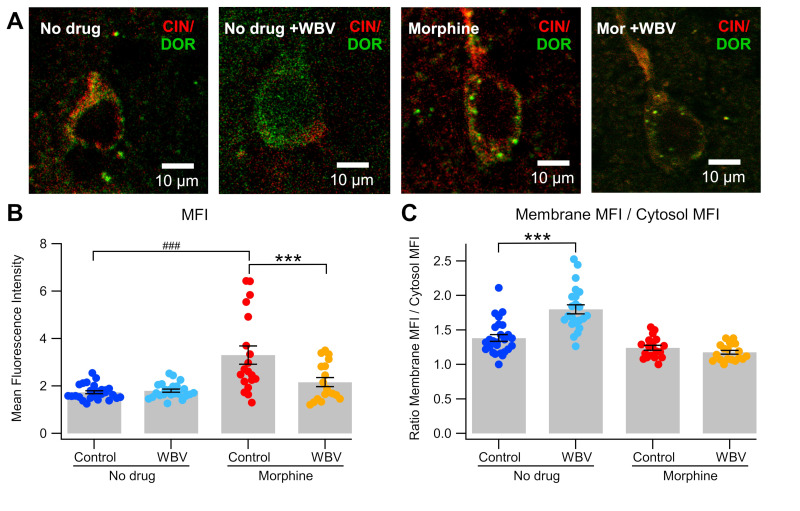
Effects of WBV on DOR expression in CINs. (**A**) This panel of representative immunohistochemical images shows high magnification views (40×) of DOR expression in CINs with no drug, + WBV, sub-chronic morphine, and sub-chronic morphine + WBV conditions. (**B**) WBV significantly decreased the enhancement of DOR expression produced by sub-chronic morphine. (**C**) WBV significantly increased membrane translocation of DORs, as reported previously, but morphine was without effect on translocation. Significance levels for comparisons within a drug condition are indicated with asterisks (***, *p* < 0.001) and for between drug conditions with hashes (###, *p* < 0.001). Not all comparisons are included.

**Figure 4 ijms-24-14147-f004:**
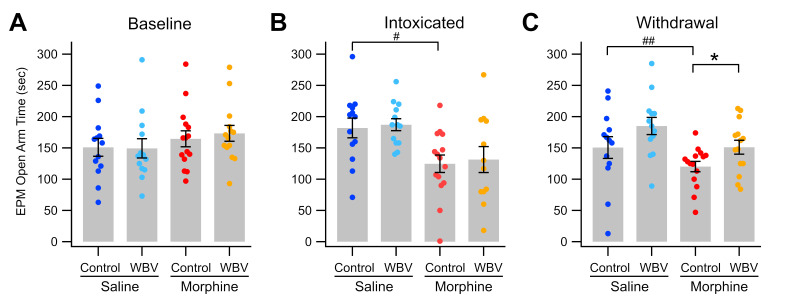
Effects of WBV on anxiety in the elevated plus maze procedure. Twenty-four hours following the last morphine injection, rats were placed in the elevated plus maze (EPM) procedure and the time spent in the open arm was recorded. Comparisons between treatment conditions: (**A**) baseline; (**B**) during morphine intoxication; and (**C**) during morphine withdrawal. Note that there were no differences in open arm time in baseline, but that sub-chronic morphine significantly reduced the time in the open arm, and that WBV significantly reduced the anxiety associated with morphine withdrawal. Significance levels for comparisons within a drug condition are indicated with asterisks (*, *p* < 0.05) and for between drug conditions with hashes (#, *p* < 0.05; ##, *p* < 0.01) Not all comparisons are included.

## Data Availability

The data supporting the findings of this study are available from the corresponding author upon reasonable request.

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
