# Peer review of "Whole-Body Vibration Prevents Neuronal, Neurochemical, and Behavioral Effects of Morphine Withdrawal in a Rat Model"

_ijms, 2023, doi:10.3390/ijms241814147_

Round 1

Reviewer 1 Report

Review:

Whole-Body Vibration Prevents Neuronal, Neurochemical, and  Behavioral Effects of Chronic Morphine Withdrawal in a Rat  Model

This is an interesting study with  whole-body vibration (WBV) to help with potentially a number of issues such as ETOH and  opioid withdrawal. This study in rodents could progress to human studies quickly as it appears to be non-invasive. How WBV blocks the desensitization of VTA GABA neurons to reinstatement exposure is really a key to the mechanisms to understand. The study is well explained and the methods are well detailed.

Line 124 “As WBV has previously been shown to alter chronic EtOH-induced changes to NAc  DA release” It would help the reader to just know “decrease “ or “increase “ as compared to just “altered”.

Line 142.” Taken altogether, these results suggest that WBV blocks chronic morphine-induced changes to NAc DA neural activity. “ Is it reasonable to maybe also assume that WBV induces a release of endorphins and then decreases the sensitivity to exogenously given morphine?

It is known that exercise and acupuncture results in the release of endorphins. Thus, it is not surprising it helps to reduce, replace and dampen withdrawals. So, would WBV potentially be acting in the same manner ?

 I am not familiar, but the topic  in the manuscript is aimed at the same point that endorphins are released with physical exercise and acupuncture, so is it known if dopamine also increases with exercise and acupuncture. It would help a reader to pull together the generally known concepts to the effects of WBV.

These comments are just suggestions for the authors to think about and see if they want to address in the manuscript.

Author Response

Reviewer 1 Comment 1:

Line 124 “As WBV has previously been shown to alter chronic EtOH-induced changes to NAc  DA release” It would help the reader to just know “decrease “ or “increase “ as compared to just “altered”.

Response 1:

Line 124 word “altered” changed to “decrease” to clarify.

Reviewer 1 Comment 2: 

Line 142.” Taken altogether, these results suggest that WBV blocks chronic morphine-induced changes to NAc DA neural activity. “ Is it reasonable to maybe also assume that WBV induces a release of endorphins and then decreases the sensitivity to exogenously given morphine?

It is known that exercise and acupuncture results in the release of endorphins. Thus, it is not surprising it helps to reduce, replace and dampen withdrawals. So, would WBV potentially be acting in the same manner ?

 I am not familiar, but the topic  in the manuscript is aimed at the same point that endorphins are released with physical exercise and acupuncture, so is it known if dopamine also increases with exercise and acupuncture. It would help a reader to pull together the generally known concepts to the effects of WBV

Response 2:

Clarification added at line 285 as follows:  It is noteworthy that WBV shares potential mechanistic underpinnings with several physical therapeutics including osteopathic and chiropractic manipulation and acupuncture. Future studies should evaluate effectiveness in a human model and could include studies into mesolimbic effects of these varied physical modalities.

Reviewer 2 Report

1. The Authors induced spontaneous withdrawal syndrome after 5 days of the treatment. The period is fine, however the Authors  should indicate the symptoms on the basis of which it was determined that the animals have withdrawal syndrome, such as wet dog shaking, stereotyped movements, standing,  jumping, tooth tremors and chewing, or runny nos, etc.

2. As there were 83 animals and four groups were used in the study, please provide the numebr of animals per group. Please, also consider the division of animals by the type of the study.

3. Please type the sex of the animals used.

4. Please provide on what basis did the Authors choose to use the exact dose of morphine while 80 Hz WBV? Was there any dose-response analsis conducted?

5. Morphine given subcutaneously for 5 days creates rather a subchronic morphine exposure model, not chronic.

minor changes are required

Author Response

Reviewer 2 Comment 1:

The Authors induced spontaneous withdrawal syndrome after 5 days of the treatment. The period is fine, however the Authors  should indicate the symptoms on the basis of which it was determined that the animals have withdrawal syndrome, such as wet dog shaking, stereotyped movements, standing,  jumping, tooth tremors and chewing, or runny nos, etc.

Response 1:

The following line and reference were added in methods section 4.3 at line 324: “Evaluative signs of withdrawal included wet dog shakes, teeth chattering, mastication, and eye twitch (Maldonado & Koob, 1993).”

Reviewer 2 Comment 2:

As there were 83 animals and four groups were used in the study, please provide the number of animals per group. Please, also consider the division of animals by the type of the study.

Response 2:

Line 314 of section 4.3 has been modified to say: “Rats were randomly divided into four groups based on drug administered and treatment condition as follows: saline + no treatment, n=20; saline + 80 Hz WBV, n=20; morphine + no treatment, n=22; morphine + 80 Hz WBV, n=21.” Additionally, sample sizes for other experiments were added in the methods section where applicable.

Reviewer 2 Comment 3:

Please type the sex of the animals used.

Response 3:

Sex of animals used has been added at line 292 with justification for rationale as follows: “Initial testing with male and female rats demonstrated no significant difference in WBV-induced effects on mesolimbic neurons.”

Reviewer 2 Comment 4:

Please provide on what basis did the Authors choose to use the exact dose of morphine while 80 Hz WBV? Was there any dose-response analsis conducted?

Response 4:

The selection of vibration at a frequency of 80 Hz is justified in the introduction, lines 72-82. For further clarification, the following line and references were also added to methods section 4.2 line 310: “These parameters were selected based upon the results of prior studies (Bills et al., 2020; Bills et al., 2022).”  

Reviewer 2 Comment 5:

Morphine given subcutaneously for 5 days creates rather a subchronic morphine exposure model, not chronic.

Response 5:

Thank you for your comment about the exposure paradigms. The point is well taken that there are many chronic models for the induction of morphine dependence and the definition of chronic and sub-chronic exposure should be the focus of further clarification.  Given that there is precedence for the description of a variety of paradigms as chronic we agree that broader clarification and consensus is needed on this issue to better define the meaning of chronic.  We suggest that this issue is beyond the scope of this paper.